# Skewness Preference and Asset Pricing: Evidence from the Japanese Stock Market

**Sheng-Ping Yang \*** and **Thanh Nguyen**

Department of Economics and Management, Gustavus Adolphus College, 800 W. College Ave., Saint Peter, MN 56082, USA
\* Correspondence: syang10@gustavus.edu; Tel.: +1-507-933-7014; Fax: +1-507-933-6032

**Abstract:** Previous studies have shown that investor preference for positive skewness creates a potential premium on negatively skewed assets. In this paper, we attempt to explore the connection between investors' skewness preferences and corresponding demand for a risk premium on asset returns. Using data from the Japanese stock market, we empirically study the significance of risk aversion with skewness preference that potentially delivers a premium. Compared to studies on other stock markets, our finding suggests that Japanese investors exhibit preference for positively skewed assets, but do not display dislike for ones that are negatively skewed. This implies that investors from different countries having dissimilar attitudes toward risk may possess different preferences toward positive skewness, which would result in a different magnitude of expected risk premium on negatively skewed assets.

**Keywords:** asset pricing; higher moment estimators; Japanese stock market; risk preferences; systematic skewness

**JEL Classification:** C33; G11; G12

---

## 1. Introduction

The traditional mean-variance framework of an asset pricing model has been used widely by both academia and practitioners in the last four decades. The approaches using parametric statistics rely largely on either the assumption of investors' quadratic utility functions or the normal distribution of securities returns (Sharpe 1964; Lintner 1965). Despite extensive popularity, asset pricing models have been questioned on not fully considering investor risk preference, and, as a result, the empirical studies poorly explain the cross-sectional variation in asset returns, as there may be other factors, such as market capitalization and book-to-market equity ratio, that may potentially affect the stock prices (Fama and French 2004; Jegadeesh and Titman 1993). Consequently, the assumptions of normality in asset return distribution as well as the quadratic utility functions are considered impractical.[1]

The method of stochastic dominance (SD) developed by Hadar and Russell (1969), Hanoch and Levy (1969), Rothschild and Stiglitz (1970), and Whitmore (1970) is an alternative way to the mean-variance approach. This method is a non-parametric approach, which considers the entire distribution of returns rather than just the first two moments (Kuosmanen 2001). Since it requires less restrictive assumptions about the investors' utility functions and relies on general assumptions about investor non-satiety

---

[1] Beedles (1979) and Schwert (1989) show the presence of both non-normality and skewness in asset return distribution. In addition, Maggi (2004) and Post and Levy (2005) agree that an inverse S-shaped utility function better explains investors' loss aversion.

and risk preferences, the SD approach provides practitioners a powerful tool to rank different assets, indices, and individual stocks as well as to form portfolios without subject to much restriction.

There are three main orders of stochastic dominance. While the first order implies that the utility function is increasing and investors prefer more wealth to less, the second order goes beyond the level of the first order considering that investors are risk averse and their utility functions are increasingly concave. The third order covers additional ranking of portfolios to investors' preference for positive skewness and dislike for negative skewness. In this paper, we attempt to show rationality that affects investor portfolio choices based on third-order stochastic dominance.

The hypothesis of the study is that Japanese investors exhibit preference for positive skewness by forming portfolios of third-order stochastic dominant and dominated stocks. If they have preferences for assets that are positively skewed, longing portfolios of dominant stocks and shorting portfolios of dominated stocks should yield a premium over the market. In fact, Kassimatis (2011) shows that these type of portfolios on average generate 2.89% per month in the U.K. market over the period 1998–2009. Our reason for choosing the Japanese market is because (i) the Japanese market is one of the most developed markets in the world, (ii) Japanese investors, as influenced by the Asian culture, may exhibit different investing preference, and (iii) the Japanese economy has experienced a very slow growth rate since the 1990s, which may cause Japanese investors to present different behavior from their peers in the U.S. and U.K. markets.

The paper is organized as follows: Section 2 discusses the literature review of pricing skewness in financial markets and the Japanese stock market, Section 3 describes data and methodology, Section 4 presents the results, and Section 5 is the conclusion.

## 2. Literature Review

### 2.1. Pricing Skewness in Financial Markets

Evidence on the effect of skewness in asset pricing has been relatively consistent as investors are prone to select securities or portfolios that are right-skewed and demand a risk premium on those that are left-skewed. Empirical studies mostly support the skewness preference implied by third-order stochastic dominance, such as those by Arditti (1967), Kraus and Litzenberger (1976), Friend and Westerfield (1980), Harvey and Siddique (2000), Smith (2007), and Doan et al. (2010). In their approach to analyze market portfolio efficiency relative to benchmark portfolios, Post and Levy (2005) find in their sample that the high yield portfolio of stocks has a negatively skewed return distribution compared to the low yield portfolio with a positively skewed return distribution.

A recent study by Blau (2017) presents strong evidence that a risk premium on stock return for stocks with negative skewness is driven primarily by high investor sentiment. Using a stochastic dominance approach, Fong (2013) also studies investor sentiment and risk preferences on so-called lottery stocks and finds that investor sentiment does explain the demand and returns of lottery stocks. These findings support an earlier study by Wen and Wen and Yang (2009) who suggest that pricing skewness of asset return is highly correlated with speculative sentiment in the market. Using samples of 33 composite indexes, they found evidence that investors demand only risk compensation in the market where return distribution is negatively skewed.

### 2.2. Overview of the Japanese Stock Market

With its size of the economy, Japan has remained as one of the key players in the global economic system, accounting for about 4 percent of the world's GDP. Its stock market with capitalization of more than $5 trillion is the third largest in the world after the Chinese and the U.S. stock markets. Moreover, the Japanese stock market is considered one of the most efficient markets in the world along with the U.S. and the U.K. markets. Nevertheless, since the early 2000s, traders have perceived significant intervention by the Bank of Japan in the foreign exchange market, causing a substantial effect on the stock market. Akiba et al. (2013) applied a microstructural model to study the impact of the Bank of Japan

intervention in the yen/dollar foreign exchange market. The empirical results revealed a significant increase in the number of informed traders after a series of interventions. Also using a framework of a microstructure trade model to directly measure traders' beliefs, Gençay and Gradojevic (2013) analyzed the risk of informed trading in foreign exchange markets and found that Japanese traders were better informed and thus dominant on the market. These are consistent with an early study by Covrig and Melvin (2002) who found Japanese quotes tend to lead the rest of the market when the informed are active.

Despite its prevailing role in the world economy, Japan's economy has been stagnant with a very low growth rate for the last 20 years. Some indicators of this stagnancy include a non-growing labor force, rising unemployment, decreasing saving rates, and near-to-zero interest rates. Because of the economic stagnancy yet still highly efficient stock market, we expect to yield results that are different from what has been tested in other stock markets.

As the asset pricing model has been criticized for its unrealistic assumptions, empirical studies in Japanese stocks are not immune to these issues. Walid and Ahlem (2009) suggested that the capital asset pricing model (CAPM) is not appropriate in the Japanese stock market by testing the market during the period 2002–2007. Motivated by the high book-to-market premium in Japanese stocks, Daniel et al. (2001) examined the value premium of Japanese stock return over the period 1975–1997 and the results reject the Fama–French three-factor model. A more recent study by Bretschger and Bretschger and Lechthaler (2012) also observe that the CAPM model does not hold true in the Japanese stock market.

There have been several studies using the SD method on the Japanese stock market. Cho et al. (2007), using the SD criteria, suggested that there was a Monday effect for the Nikkei 225 over the period 1990–2004. Fong et al. (2005) confirmed that the winner portfolios stochastically dominated loser portfolios at the second and third orders over the period 1989–2001. Furthermore, Lean et al. (2007) indicated that there was no January effect in the Japanese market.[2] In this study, the January returns do not appear to have any predictive power for the rest of the year by first-, second-, or third-order stochastic dominance. In the same study, Lean shows that Monday stock returns are stochastically dominated by Tuesday returns at the first order.

## 3. Data and Methodology

The data applied in this study are rates of return of 197 stocks drawn from the Nikkei 225 index spanning from May 2004 to October 2014 extracted from Datastream. Japan's Nikkei 225 Average is the leading index of Japanese stocks, with which the market value of the index covers more than half the market capitalization of the Japanese stock exchange. We exclude those stocks where trading is not very active or discontinues throughout the estimation period and those that are subject to various types of changes. In the late 1990s, Japan was enduring a long economic recession causing the stock market to be in several rounds of a bear market. The market started seeing a revival in the second half of 2003 and the bullish trend continued until 2007 (Shibata 2012).[3] For this reason, we chose to select data beginning in 2004.

To identify the dominant and dominated stocks, we first estimated the distribution of daily returns for each stock relative to the stock index returns for the six-month period prior to the month, and the last five observations of the previous month were excluded. This was applied to each of the 84 months in the sample. We assumed that investors consider the return distribution of a stock only from the most recent history of the stock. A six-month period of daily returns is sufficiently long to provide a full distribution of returns.

---

2　　The January effect refers to the fact that common stock returns are larger in January than other months.
3　　Following the global financial crisis in 2007, Japanese stocks tumble into a bear market (Kawai and Takagi 2009).

Table 1 reports the statistics on the number of dominant and dominated stocks in terms of the stock index in every six-month period. Cleary, the number of dominated stocks is much larger than the number of dominant stocks. The stock index is the most diversified portfolio in the sample, so the expectation is that most stocks are dominated by the index, and the results confirm this expectation. On average, 85.5 stocks are dominated by the stock index and only 7.25 stocks dominate the stock index in the sample. We find that the highest number of dominated stocks for the six-month period occurs in the period of September 2009 to February 2010, with 125 stocks being dominated by the index. The highest number of dominant stocks occurs for the six-month period ending September 2010, with 26 stocks dominating the index.

To construct the dominant and dominated portfolios for each of the selected bearish and bullish markets, we selected a period for the bear market from June 2007 to August 2008 and a period for the bull market from April to September 2005.[4] That is, one dominant portfolio and one dominated portfolio were constructed in respective bearish and bullish periods. Since the number of dominated stocks in each six-month period was relatively large, we randomly selected 18 dominated stocks to form a dominated portfolio for each market. Tables 2 and 3 report the statistics of the Sharpe ratios and Sortino ratios for dominant and dominated portfolios in different states of markets. The Sharpe and Sortino ratios of the market index are also included for comparison. If the Sortino ratio is more than the Sharpe ratio, it signifies that the asset is more likely to have a positively skewed return. On the other hand, a negatively skewed return is more likely when the Sortino ratio is less than the Sharpe ratio.[5] Both the dominant and dominated portfolios appear to be mostly positively skewed during the bull market and negatively skewed during the bearish market.

Stock return analysis based on the SD rules is pertaining to the assumption that investors' reactions differ in response to potential gains and losses (Kassimatis 2011). There are several orders of stochastic dominance, with which orders of stochastic dominance can increase with higher order derivatives applied to the utility function. First-order stochastic dominance (FSD) implies that utility functions exhibit non-satiation, where more is preferred to less; second-order stochastic dominance (SSD) requires prevailing risk aversion sentiment in addition to non-satiation preference; and, third-order stochastic dominance (TSD) requires non-satiation, risk aversion, and preference for positive skewness (Post and Levy 2005; Al-Khazali et al. 2014; Kassimatis 2011).

The principle of expected utility maximization is consistent with the SD rules in the sense that preferences that follow *n*-order SD are equivalent to preferences on *n*-order risk averters (Li and Wong 1999). It implies that the expected utilities of investors are higher for those who hold portfolios of dominant stocks than the dominated ones. More specifically, third-order stochastic dominance, as we focus on in this paper, implies an investor's preference for positive skewness and dislike for negative skewness.

The method we applied to identify the dominant and dominated stocks follows the algorithms of stochastic dominance proposed by Babbel and Herce (2007). Suppose there are two assets *X* and *Y* with probability of any return in *X* being always at least as high as in *Y*. An investor who is non-satiated will prefer asset *X* to asset *Y*. The degree of stochastic dominance generally delineates the determination of an order of preference between two assets rather than the assumptions of return distribution and risk assessment. Consequently, an investment decision can be described even without a specific form of the investor's utility function. In such a case, *X* can be denoted as the returns of *W* with corresponding cumulative distribution function (CDF) *F*, while *Y* as the returns of L with CDF *G*. Rational investors who want to maximize their expected utility would prefer *F* to *G*. That is, *X* always generates better chances than *Y* for investors to earn higher returns regardless of investors' preferences for risk.

---

[4]　Shibata (2012) investigated the duration of bull and bear markets using the Tokyo Stock Price Index as proxy over the period 1949–2007. We selected the bear and bull markets within the estimation period for this study.

[5]　Note both the Sharpe and Sortino ratios are only meaningful tools of analysis if the ratios are positive and the distribution of performance of the underlying investments approximately resembles a normal distribution.

**Table 1.** The number of dominated and dominant stocks in each six-month sub-period.

| Period (Year: Month–Year: Month) | Dominated Stock | Dominant Stock | Period (Year: Month–Year: Month) | Dominated Stock | Dominant Stock |
|---|---|---|---|---|---|
| 2004: 5–2004: 10 | 68 | 20 | 2009: 5–2009: 10 | 103 | 1 |
| 2004: 6–2004: 11 | 83 | 4 | 2009: 6–2009: 11 | 123 | 1 |
| 2004: 7–2004: 12 | 85 | 5 | 2009: 7–2009: 12 | 119 | 1 |
| 2004: 8–2005: 1 | 71 | 8 | 2009: 8–2010: 1 | 110 | 2 |
| 2004: 9–2005: 2 | 78 | 0 | 2009: 9–2010: 2 | 125 | 2 |
| 2004: 10–2005: 3 | 72 | 2 | 2009: 10–2010: 3 | 102 | 1 |
| 2004: 11–2005: 4 | 44 | 6 | 2009: 11–2010: 4 | 76 | 4 |
| 2004: 12–2005: 5 | 50 | 4 | 2009: 12–2010: 5 | 66 | 3 |
| 2005: 1–2005: 6 | 44 | 4 | 2010: 1–2010: 6 | 71 | 21 |
| 2005: 2–2005: 7 | 60 | 4 | 2010: 2–2010: 7 | 84 | 23 |
| 2005: 3–2005: 8 | 59 | 2 | 2010: 3–2010: 8 | 75 | 21 |
| 2005: 4–2005: 9 | 59 | 1 | 2010: 4–2010: 9 | 86 | 26 |
| 2005: 5–2005: 10 | 81 | 0 | 2010: 5–2010: 10 | 89 | 24 |
| 2005: 6–2005: 11 | 83 | 0 | 2010: 6–2010: 11 | 102 | 10 |
| 2005: 7–2005: 12 | 81 | 0 | 2010: 7–2010: 12 | 83 | 0 |
| 2005: 8–2006: 1 | 74 | 0 | 2010: 8–2011: 1 | 74 | 2 |
| 2005: 9–2006: 2 | 74 | 1 | 2010: 9–2011: 2 | 94 | 0 |
| 2005: 10–2006: 3 | 76 | 2 | 2010: 10–2011: 3 | 74 | 14 |
| 2005: 11–2006: 4 | 95 | 3 | 2010: 11–2011: 4 | 79 | 10 |
| 2005: 12–2006: 5 | 87 | 3 | 2010: 12–2011: 5 | 72 | 11 |
| 2006: 1–2005: 6 | 82 | 8 | 2011: 1–2011: 6 | 86 | 10 |
| 2006: 2–2006: 7 | 94 | 8 | 2011: 2–2011: 7 | 90 | 9 |
| 2006: 3–2006: 8 | 84 | 8 | 2011: 3–2011: 8 | 83 | 15 |
| 2006: 4–2006: 9 | 94 | 10 | 2011: 4–2011: 9 | 101 | 20 |
| 2006: 5–2006: 10 | 95 | 9 | 2011: 5–2011: 10 | 97 | 19 |
| 2006: 6–2006: 11 | 94 | 4 | 2011: 6–2011: 11 | 92 | 16 |
| 2006: 7–2006: 12 | 109 | 1 | 2011: 7–2011: 12 | 89 | 19 |
| 2006: 8–2007: 1 | 105 | 0 | 2011: 8–2012: 1 | 89 | 18 |
| 2006: 9–2007: 2 | 94 | 0 | 2011: 9–2012: 2 | 99 | 9 |
| 2006: 10–2007: 3 | 73 | 0 | 2011: 10–2012: 3 | 100 | 2 |
| 2006: 11–2007: 4 | 70 | 0 | 2011: 11–2012: 4 | 107 | 2 |
| 2006: 12–2007: 5 | 83 | 0 | 2011: 12–2012: 5 | 99 | 5 |
| 2007: 1–2007: 6 | 84 | 0 | 2012: 1–2012: 6 | 116 | 3 |
| 2007: 2–2007: 7 | 92 | 0 | 2012: 2–2012: 7 | 116 | 14 |
| 2007: 3–2007: 8 | 74 | 2 | 2012: 3–2012: 8 | 114 | 17 |
| 2007: 4–2007: 9 | 75 | 3 | 2012: 4–2012: 9 | 104 | 18 |
| 2007: 5–2007: 10 | 80 | 2 | 2012: 5–2012: 10 | 102 | 16 |
| 2007: 6–2007: 11 | 90 | 10 | 2012: 6–2012: 11 | 104 | 3 |
| 2007: 7–2007: 12 | 97 | 11 | 2012: 7–201: 12 | 91 | 0 |
| 2007: 8–2008: 1 | 109 | 17 | 2012: 8–2013: 1 | 67 | 0 |
| 2007: 9–2008: 2 | 99 | 20 | 2012: 9–2013: 2 | 73 | 1 |
| 2007: 10–2008: 3 | 99 | 20 | 2012: 10–2013: 3 | 98 | 4 |
| 2007: 11–2008: 4 | 104 | 20 | 2012: 11–2013: 4 | 52 | 0 |
| 2007: 12–2008: 5 | 90 | 9 | 2012: 12–2013: 5 | 76 | 1 |
| 2008: 1–2008: 6 | 86 | 5 | 2013: 1–2013: 6 | 69 | 6 |
| 2008: 2–2008: 7 | 95 | 5 | 2013: 2–2013: 7 | 66 | 3 |
| 2008: 3–2008: 8 | 99 | 11 | 2013: 3–2013: 8 | 61 | 4 |
| 2008: 4–2008: 9 | 91 | 6 | 2013: 4–2013: 9 | 59 | 1 |
| 2008: 5–2008: 10 | 96 | 17 | 2013: 5–2013: 10 | 53 | 12 |
| 2008: 6–2008: 11 | 104 | 17 | 2013: 6–2013: 11 | 87 | 8 |
| 2008: 7–2008: 12 | 109 | 16 | 2013: 7–2013: 12 | 57 | 1 |
| 2008: 8–2009: 1 | 100 | 18 | 2013: 8–2014: 1 | 81 | 6 |
| 2008: 9–2009: 2 | 103 | 17 | 2013: 9–2014: 2 | 105 | 3 |
| 2008: 10–2009: 3 | 98 | 11 | 2013: 10–2014: 3 | 94 | 2 |
| 2008: 11–2009: 4 | 64 | 5 | 2013: 11–2014: 4 | 92 | 5 |
| 2008: 12–2009: 5 | 59 | 2 | 2013: 12–2014: 5 | 78 | 11 |
| 2009: 1–2009: 6 | 66 | 1 | 2014: 1–2014: 6 | 79 | 10 |
| 2009: 2–2009: 7 | 69 | 0 | 2014: 2–2014: 7 | 80 | 11 |
| 2009: 3–2009: 8 | 62 | 2 | 2014: 3–2014: 8 | 74 | 13 |
| 2009: 4–2009: 9 | 83 | 1 | 2014: 4–2014: 9 | 89 | 11 |

**Table 2.** The statistics of Sharpe and Sortino ratios for a dominant portfolio.

| Period after | 1st Month | 2nd Month | 3rd Month | 4th Month | 5th Month | 6th Month |
|---|---|---|---|---|---|---|
| Overall Market | | | | | | |
| Dominant Portfolio | | | | | | |
| Sharpe ratio | −0.06 | −0.12 | 0.04 | 0.12 | −0.05 | 0.02 |
| Sortino ratio | −0.08 | −0.16 | 0.05 | 0.17 | −0.07 | 0.03 |
| Market Index | | | | | | |
| Sharpe ratio | 0.03 | 0.02 | 0.04 | 0.07 | 0.03 | 0.12 |
| Sortino ratio | 0.04 | 0.02 | 0.05 | 0.08 | 0.04 | 0.14 |
| Bull Market | | | | | | |
| Dominant Portfolio | | | | | | |
| Sharpe ratio | 1.09 | 1.17 | 0.63 | 0.58 | 0.85 | 0.24 |
| Sortino ratio | 1.92 | 17.98 | 4.35 | 3.27 | 32.03 | 0.46 |
| Market Index | | | | | | |
| Sharpe ratio | 1.20 | 1.29 | 1.50 | 1.57 | 0.90 | 0.85 |
| Sortino ratio | 2.98 | 3.26 | 2.73 | 2.68 | 1.48 | 1.27 |
| Bear Market | | | | | | |
| Dominant Portfolio | | | | | | |
| Sharpe ratio | −0.14 | −0.47 | −0.37 | −0.39 | −0.36 | −0.33 |
| Sortino ratio | −0.21 | −0.61 | −0.48 | −0.38 | −0.46 | −0.54 |
| Market Index | | | | | | |
| Sharpe ratio | −0.45 | −0.56 | −0.56 | −0.46 | −0.52 | −0.48 |
| Sortino ratio | −0.79 | −0.65 | −0.65 | −0.53 | −0.66 | −0.55 |

**Table 3.** The statistics of Sharpe and Sortino ratios for a dominated portfolio.

| Period after | 1st Month | 2nd Month | 3rd Month | 4th Month | 5th Month | 6th Month |
|---|---|---|---|---|---|---|
| Overall Market | | | | | | |
| Dominated Portfolio | | | | | | |
| Sharpe ratio | −0.01 | −0.06 | −0.07 | −0.03 | −0.12 | −0.18 |
| Sortino ratio | −0.01 | −0.07 | −0.10 | −0.04 | −0.15 | −0.18 |
| Market Index | | | | | | |
| Sharpe ratio | −0.01 | −0.03 | −0.04 | −0.04 | −0.05 | −0.04 |
| Sortino ratio | −0.01 | −0.04 | −0.04 | −0.04 | −0.06 | −0.04 |
| Bull Market | | | | | | |
| Dominated Portfolio | | | | | | |
| Sharpe ratio | 1.51 | 0.86 | 1.23 | 1.37 | 0.88 | 0.54 |
| Sortino ratio | 4.08 | 2.70 | 2.85 | 2.62 | 4.66 | 1.45 |
| Market Index | | | | | | |
| Sharpe ratio | 1.20 | 1.29 | 1.50 | 1.57 | 0.90 | 0.85 |
| Sortino ratio | 2.98 | 2.79 | 2.73 | 2.68 | 1.48 | 1.27 |
| Bear Market | | | | | | |
| Dominated Portfolio | | | | | | |
| Sharpe ratio | −0.43 | −0.50 | −0.55 | −0.45 | −0.46 | −0.57 |
| Sortino ratio | −0.60 | −0.56 | −0.75 | −0.56 | −0.44 | −0.46 |
| Market Index | | | | | | |
| Sharpe ratio | −0.52 | −0.53 | −0.50 | −0.48 | −0.55 | −0.54 |
| Sortino ratio | −0.81 | −0.59 | −0.56 | −0.54 | −0.67 | −0.65 |

The three basic SD rules are (1) asset $X$ with a CDF $F$ dominates asset $Y$ with CDF $G$ by first-order SD if and only if $F_1(x) \leq G_1(x)$ for all possible returns x; (2) asset X dominates asset Y by second-order SD if and only if $F_2(x) \leq G_2(x)$ for all possible returns $x$, and $F_2$ and $G_2$ are the areas under $F$ and $G$, respectively; and (3) asset $X$ dominates asset $Y$ by third-order SD if and only if $\mu_F \geq \mu_G$ and $F_3(x) \leq G_3(x)$ for all possible returns $x$, and $F_3$ and $G_3$ are the areas under $F_2$ and $G_2$ respectively.

Let $U$ be the investor's utility function. For FSD, non-satiation is represented by an increasing utility ($U' \geq 0$). For SSD, non-satiation and risk aversion are represented by an increasingly concave utility function ($U' \geq 0$ and $U'' \leq 0$). Additionally, TSD requires $U' \geq 0$, $U'' \leq 0$, and $U''' \geq 0$, where the utility function is increasingly concave over gains and increasingly convex over losses. Accordingly, there is apparently a hierarchical relationship as FSD implies SSD, which in turn implies TSD (Levy 1992).

## 4. Estimation Results

Table 4 presents the performance of both the dominant and dominated portfolios over the entire sample period. Both of the portfolios appear to have negative average monthly returns for each of the following six months after they are formed. We further compared each of the monthly returns to the Nikkei 225 index of the same month by taking the difference between the return of the dominant/dominated portfolios and the index. A negative value indicates that the portfolio performs worse than the market. The results show that both the portfolios were outperformed by the market during the sample period, with which the dominant portfolio appears to perform better than the dominated portfolio during the time.

**Table 4.** Dominant and dominated portfolio returns after formation.

| Period | 1st Month | 2nd Month | 3rd Month | 4th Month | 5th Month | 6th Month |
|---|---|---|---|---|---|---|
| Dominant Portfolio | −0.29% (4.27%) | −0.54% (4.54%) | −0.29% (4.18%) | −0.41% (4.44%) | −1.09% (4.99%) | −0.36% (4.77%) |
| Difference from the market index | −0.08% (5.27%) | −0.07% (4.78%) | −0.08% (4.90%) | −0.47% (4.71%) | −0.74% (4.79%) | −0.63% (5.08) |
| Dominated Portfolio | −0.08% (7.01%) | −0.46% (7.09%) | −0.43% (6.01%) | −0.19% (6.38%) | −0.93% (7.62%) | −1.28% (7.32%) |
| Difference from the market index | −0.05% (2.44%) | −0.24% (2.80%) | −0.19% (3.27%) | 0.05% (2.84%) | −0.62% (3.10%) | −1.05% (3.24%) |

*Note:* The percentages represent the average monthly returns for each of the following six months after the portfolios are formed.

As there is a possible structural break over the entire sample period, the estimation results are subject to potential bias. As such, we further identified a subsample period over the bullish period and a subsample period over the bearish period, respectively. Table 5 presents the results of the dominant and dominated portfolios over the bull market period (April to September 2005). During this bull market, the dominant portfolio performed worse than the dominated portfolio. Specifically, upon the first month after the dominant portfolio is formed, the portfolio yields only 2.32% return while the dominated portfolio yields 3.13%. The gap between the returns of the two portfolios increases over time until the sixth month. When compared to the Nikkei index, both the portfolios were outperformed by the market.

**Table 5.** Bullish period (2005: 4–2005: 9) returns after formation.

| Period | 1st Month | 2nd Month | 3rd Month | 4th Month | 5th Month | 6th Month |
|---|---|---|---|---|---|---|
| Dominant Portfolio | 2.32% (2.12%) | 2.50% (2.14%) | 1.36% (2.14%) | 1.49% (2.59%) | 1.87% (2.21%) | 0.71% (3.00%) |
| Difference from the market index | −1.21% (3.03%) | −2.12% (4.47%) | −4.14% (4.42%) | −4.11% (4.84%) | −2.57% (5.98%) | −3.10% (7.01%) |
| Dominated Portfolio | 3.13% (2.07%) | 3.47% (4.02%) | 4.74% (3.86%) | 5.28% (3.86%) | 5.15% (5.85%) | 2.57% (4.76%) |
| Difference from the market index | −0.40% (2.08%) | −1.14% (0.94%) | −0.76% (2.27%) | −0.32% (3.66%) | 0.71% (2.89%) | −1.24% (2.70%) |

*Note:* The percentages represent the average monthly returns for each of the following six months after the portfolios are formed.

Table 6 presents the results for the dominant and dominated portfolios over the bear market period. Different from the result obtained from the bullish period, the dominant portfolio significantly outperformed both the dominated portfolio and the market index during this bearish market from June 2007 to August 2008. Compared to the Nikkei index, the dominated portfolio generates, on average over a six-month period, 2.74% higher return than the market index, whereas the return of the dominated portfolio seems to follow a relatively random pattern.

**Table 6.** Bearish period (2007: 8–2008: 8) returns after formation.

| Period | 1st Month | 2nd Month | 3rd Month | 4th Month | 5th Month | 6th Month |
|---|---|---|---|---|---|---|
| Dominant Portfolio | −0.73% (5.24%) | −2.19% (4.67%) | −1.41% (3.82%) | −2.15% (5.50%) | −2.29% (6.40%) | −2.00% (6.08%) |
| Difference from the market index | 2.24% (5.95%) | 2.97% (6.64%) | 3.79% (6.55%) | 2.24% (5.69%) | 2.71% (6.23%) | 2.50% (7.28%) |
| Dominated Portfolio | −2.97% (6.61%) | −5.16% (9.27%) | −5.20% (9.25%) | −4.39% (9.58%) | −5.00% (9.69%) | −4.50% (9.47%) |
| Difference from the market index | 0.01% (2.28%) | −1.11% (4.24%) | −0.06% (5.14%) | 0.65% (2.56%) | −0.24% (4.54%) | −1.77% (4.32%) |

*Note:* The percentages represent the average monthly returns for each of the following six months after the portfolios are formed.

Overall, the dominant portfolio performed better than the market index over the selected bearish period but worse over the selected bullish period. The estimation results imply that Japanese investors seem to exhibit little or no preference for positive skewness in bullish market conditions. However, during a bear market, Japanese investors show preference for dominant stocks, which have a shorter left tail. This tendency may imply that Japanese investors are relatively defensive, and they tend to choose investments that have lower potential loss during a bear market.

## 5. Conclusions

The purpose of our study was to examine the significance of positive skewness on asset pricing and to test whether Japanese investors exhibit a preference for positive skewness. We used the data from the Nikkei 225 index to construct the dominant and dominated portfolios, in which the data were rebalanced monthly. Then, we empirically tested the performance of these portfolios comparing to the

stock index in both bearish and bullish periods. The bearish period was from June 2007 to August 2008, and the bullish period was from April to September 2005. We found that the dominant portfolio appears to perform better than the market during both the bearish and bullish markets, whereas the performance of dominated portfolios seems to be random during both the markets. In addition, the dominant portfolio tends to outperform the market during the bearish market rather than during the bullish market.

Our finding suggests that Japanese investors exhibit a preference for positive skewness during a bearish market, but do not display dislike for negative skewness during a bullish market. Preference for positive skewness seems to be slightly significant in asset pricing. Moreover, the fact that the dominant portfolios performed better during the bearish market implies that Japanese investors are relatively more defensive and reveal a pattern of more risk aversion. Our results also raise a question on whether an inverse S-shaped utility function captures the Japanese investors' behavior well because the dominated portfolios, which imply dislike for negative skewness, seem to have random performance.

**Author Contributions:** Conceptualization, S.-P.Y.; Investigation, S.-P.Y. and T.N.; Methodology, S.-P.Y. and T.N.; Writing-Original Draft Preparation, S.-P.Y. and T.N.; Writing-Review & Editing, S.-P.Y.

**Funding:** This research received no external funding.

**Conflicts of Interest:** The authors declare no conflict of interest.

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
