# Peer review of "Skewness Preference and Asset Pricing: Evidence from the Japanese Stock Market"

_jrfm, doi:10.3390/jrfm12030149_

Round 1

Reviewer 1 Report

Referee report on: “Implication of Preference for Positive Skewness to Asset Pricing: Evidence from the Japanese Stock Market” (jrfm-542155)

This is a fine paper that requires a moderate revision to become publishable in JRFM.

COMMENTS ON PAPER

1.      The paper’s methodology is straight from Babbel and Herce (2018, 10.3390/risks6010012). Thus, no methodological novelty is offered, but it is worthwhile to investigate Japanese investors’ preference for skewness in portfolio selection, mainly because the original methodology comes from credible scholars.

2.      I would like to see a better interpretation of results. In particular, what do we learn about Japanese investors? Is there anything special about the Japanese market relative to other developed markets?

3.      In regards to the methodology, in addition to the stochastic dominance analysis, please also calculate and discuss the Sharpe and Sortino ratios for all subsamples.

4.      Please have a 1-2-paragraph discussion about the potential effect of local (Japanese) private information and other market microstructure effects on your findings. Here are a few useful papers to reference:

·       Akiba, H., Kitamura, Y., Matsuda, S., & Sato, A. (2013). A microstructural effect of Japanese official intervention in the yen/dollar foreign exchange market. : Exchange Rates in Developed and Emerging Markets: Practices, Challenges and Economic Implications (pp. 59-74). Nova Science Publishers, Inc.

·       Kerry Back, Alan D Crane, Kevin Crotty, Skewness Consequences of Seeking Alpha, The Review of Financial Studies, Volume 31, Issue 12, December 2018, Pages 4720–4761.

·       Covrig, V. and Melvin, M. (2002). Asymmetric information and price discovery in the FX market: Does Tokyo know more about the Yen? Journal of Empirical Finance, 9, 271–285.

·       Gençay, Ramazan & Gradojevic, Nikola, 2013. "Private information and its origins in an electronic foreign exchange market," Economic Modelling, Elsevier, vol. 33(C), pages 86-93.

5.      I also have concerns about the data set that could be biased due to structural breaks. Please test for structural breaks and discuss how they affect your results.

6.      Although English writing is decent, a proofread of the paper by a native speaker is necessary.

Author Response

Additional paragraph is added in the section of estimation result with more detailed explanation on Japanese investors.

Tables 2 and 3 are added to include both the Sharpe and Sortino ratios for all subsamples. One paragraph on p. 6 is also added to explain these ratios.

In subsection 2.2, additional paragraph with discussion on the potential effect of private information and market microstructure effects are added.

One paragraph is added to explain the possible structure break over the entire sample period.

Reviewer 2 Report

The paper examines the relationship between skewness of return distribution and risk premium in the Japanese stock market using a relatively unusual method. I find the paper interesting, but I recommend introducing several changes.

The introduction should much stronger accentuate and describe the paper contribution. What is the novelty comparing to the remaining finance literature?

I suggest the begin the literature review with an additional separate section on pricing skewness in financial markets, and next continue with the current subsection 2.2. On the other hand, subsection 2.2 could be shortened to 1-2 paragraphs and moved to the data section 3.

The authors should more carefully explain their sample choice to assure it is free of any biases. Why the sample period is so short and ends in 2014? Why there are only 197 stocks in the sample? How were they selected? Why the authors focus only on the stocks included in the index? How do they consider the changes in the index composition?

Table 1 could be presented as a figure - I believe this would improve clarity.

Figure 1 seems redundant or should be prepared more professionally.

The distinction between bull and bear market is somewhat arbitrary? Maybe the authors could use some arbitrary rule? See for example Cooper, Gutierrez, and Hameed 2004 https://doi.org/10.1111/j.1540-6261.2004.00665.x.

Author Response

We provide a rationale on why choosing the Japanese market as the Japanese market is one of the most developed markets and also one of the most efficient market. On the other hand, Japanese investors, as influenced by the Asian culture, may exhibit different investing preference causing their demand for risk premium and preference for skewness to differ from investors in other markets.

Subsection 2.1 is revised with a new topic on pricing skewness in financial markets. The explanation of stochastic dominance approach is moved to the data section 3.

A more detailed explanation of sample choice is added to section 3 on pp. 5-6.

A more detailed discussion of how bull and bear markets are identified is added to section 3 on pp. 5-6.

Reviewer 3 Report

Dear Author(s):

The paper had addressed an interested topic, but had some issue to be clarified.

1.       English is weak. English proof is required. (Abstract to Conclusion).

2.       Abstract should be written again. Main findings and policy implications should be highlighted.

3.       The originality of this paper is not well described (Introduction part). The contribution of the existing literature on this topic should be emphasized clearly. The novelty is not anywhere in the work done.

4.       The sampling method (why Nikkei 225 index no other index). Can the results from this index generalize on all Japanese investors?

5.       Methodology used is quite weak, the first paragraph (what is the justification behind the distribution of the daily return? It’s not clear). Moreover, there is no support from literature.  

6.       Discussion is more on statistics not related to the finance theory. No discussion on how Japanese investor would benefit from this study.

Conclusion should be enhanced. More information and some policy implications

Author Response

The writing of the paper is re-checked.

Abstract is rewritten with highlight of main findings.

We readdress the rationale of choosing the Japanese stock market for the study and explain how Japanese investors may act different from those in other markets despite the market being as efficient as other major markets.

We include an additional paragraph in section 3 to explain the rationale of choosing the Nikkei 225 as proxy. The explanation of the methodology in section 3 on pp. 7-8 is extended.

Additional paragraph is added to explain the estimation result and implication of Japanese investor behavior.

Round 2

Reviewer 1 Report

I am happy with your responses.

Please make sure to correct the enumeration of sections. It is not right.

Also, it is "structural breaks", not "structure breaks".

Author Response

Both the enumeration of sections and typo on the structural break have been corrected.

Reviewer 2 Report

1. The sections are numerated incorrectly.
2. One limitation of the analyses in Tables 2 and 3 is that the negative Sharpe and Sortino ratios are not really meaningful. In the case of negative ratios, the higher value does not necessarily indicate better performance. The authors should acknowledge this limitation.
3. I strongly encourage authors to add notes below the tables to make the exhibits self-explanatory. Now, the tables not always can be easily understood without reading the article. For example, what are the numbers in brackets in Tables 4-6. I presume that p-values, but this is nowhere stated.

Author Response

All sections are correctly numerated. Footnote 5 is added to address issues of negative Sharpe and Sortino ratios. A note is added to each of Tables 4-6.